



# The response of the North Pacific jet and stratosphere-to-troposphere transport of ozone over western North America to RCP8.5 climate forcing

Dillon Elsbury[1,2], Amy H. Butler[2], John R. Albers[1,3], Melissa L, Breeden[1,3], Andrew O'Neil Langford[2]

[1]Cooperative Institute for Research in Environmental Sciences, Boulder, 80305, United States
[2]National Oceanic and Atmospheric Administration Chemical Sciences Laboratory, Boulder, 80305, United States
[3]National Oceanic and Atmospheric Administration  Physical Sciences Laboratory, Boulder, 80305, United States

*Correspondence to*: Dillon Elsbury (dillon.elsbury@noaa.gov)

**Abstract.** Stratosphere-to-troposphere transport (STT) is an important source of ozone for the troposphere, particularly over western North America. STT in this region is predominantly controlled by a combination of the variability and location of the Pacific jet stream and the amount of ozone in the lower stratosphere, two factors which are likely to change if greenhouse gas concentrations continue to increase. Here we use Whole Atmosphere Community Climate Model experiments with a tracer of stratospheric ozone (O3S) to study how end-of-the-century Representative Concentration Pathway (RCP) 8.5 sea surface temperatures (SSTs) and greenhouses gases (GHGs), in isolation and in combination, influence STT of ozone over western North America relative to a preindustrial control background state.

We find that O3S increases up to 39% at 700 hPa over western North America in response to RCP8.5 forcing with the largest increases occurring during late winter and tapering off somewhat during spring and summer. When this response is decomposed into the contributions made by future SSTs and GHGs, the latter are found to be primarily responsible for these O3S changes. Both the future SSTs and the future GHGs accelerate the Brewer Dobson circulation, which increases extratropical lower stratospheric ozone mixing ratios. While the GHGs promote a more zonally symmetric lower stratospheric ozone change due to enhanced ozone production and some transport, the SSTs increase lower stratospheric ozone predominantly over the North Pacific via transport associated with a stationary planetary-scale wave. Ozone accumulates in the trough of this anomalous wave and is reduced over the wave's ridges, illustrating that the composition of the lower stratospheric ozone reservoir in the future is dependent on the phase and position of the stationary planetary-scale wave response to future SSTs, in addition to the poleward mass transport provided by the accelerated Brewer-Dobson Circulation. In addition, the future SSTs account for most changes to the large-scale circulation in the troposphere and stratosphere compared to the effect of future greenhouse gases. These changes include modifying the position and speed of the future North Pacific jet, lifting the tropopause, accelerating both the Brewer-Dobson Circulation's shallow and deep branches, and enhancing two-way isentropic mixing in the stratosphere.






## 1 Introduction

Tropospheric ozone is a pollutant harmful to humans and vegetation, therefore understanding its response to climate change
has important implications for future air quality (Fleming et al. 2018). Future tropospheric ozone amounts are affected by
multiple processes including anthropogenic emissions and changes to the large-scale circulation, which in turn are dependent
on the choice of model and climate change scenario (Young et al. 2018). For high-end emissions scenarios (Representative
Concentration Pathway (RCP) 8.5), recent chemistry-climate models project an increase in Northern Hemisphere tropospheric
ozone (Archibald et al. 2020), largely due to enhanced methane emissions (Winterstein et al. 2019), but also due to stronger
transport of stratospheric ozone into the troposphere (Griffiths et al. 2021).

Enhanced stratosphere-to-troposphere transport (STT) of ozone is expected in the future, due in part to more frequent
tropopause folding (Akritidis et al. 2019), but also due to higher ozone mixing ratios in the lower stratosphere. Since the
amount of ozone in the lower extratropical stratospheric "reservoir," often measured on the 350 Kelvin isentrope, is positively
correlated with the amount of ozone contained in intrusions of stratospheric air exchanged into the troposphere (Albers et al.
2018), larger lower stratospheric ozone mixing ratios should coincide with more STT of ozone. A diverse set of physical and
chemical processes is anticipated to have the net effect of increasing future lower stratospheric ozone mixing ratios in the
extratropics; these processes include enhanced downwelling associated with the acceleration of the Brewer-Dobson Circulation
(Abalos et al. 2020), two-way isentropic mixing (Eichinger et al. 2019; Ball et al. 2020; Dietmüller et al. 2021), enhanced
ozone production associated with stratospheric cooling (Rind et al. 1990; Jonsson et al. 2004; Oman et al. 2010), chemical
impacts of increasing methane and nitrous oxide concentrations (Revell et al. 2012; Butler et al. 2016; Winterstein et al. 2019),
and expected emissions reductions of ozone depleting substances (ODSs) (Banerjee et al. 2016; Meul et al. 2018; Fang et al.
2019; Griffiths et al. 2020; Dietmüller et al. 2021).

While the mechanisms influencing future lower stratospheric ozone changes are fairly well established in a zonally-averaged
sense, it is less evident what role regional dynamical and chemical zonal asymmetries will play in future STT. Historically,
one of the key regions where stratospheric mass fluxes enter the lower free troposphere is over western North America
(Sprenger and Wernli 2003; Lefohn et al. 2011; Skerlak et al. 2014). Tropopause folding and STT maximize over this region
during spring, when the North Pacific jet transitions from a strong and latitudinally narrow band of westerlies to a weaker and
latitudinally broad jet (Newman and Sardeshmukh 1998; Breeden et al. 2021). Intrusions here have been observed to enhance
free tropospheric ozone concentrations beyond 30 parts per billion (Knowland et al. 2017; Langford et al. 2017; Zhang et al.
2020; Xiong et al. 2022; Langford et al. 2022). When combined with background ozone concentrations, which are also affected




by regional precursor emissions, vegetation, and upwind transport (Cooper et al. 2010; Langford et al. 2017), ozone
concentrations may exceed the surface eight-hour National Ambient Air Quality Standard (EPA 2006).

It is established that the subtropical and eddy-driven jets' response to climate change will vary by region and season (Akritidis
et al. 2019; Harvey et al. 2020). However, it is not yet known how regional jet changes, such as the spring transition of the
North Pacific jet, combined with changes to the lower stratospheric ozone reservoir, may affect STT regionally in the future.
In this study, we use a set of National Center for Atmospheric Research (NCAR) Whole Atmosphere Community Climate
Model (WACCM) experiments described in Section 2, which include fully interactive chemistry and a tracer of stratospheric
ozone (O3S), to evaluate how RCP8.5 sea surface temperatures (SSTs) and RCP8.5 greenhouse gases (GHGs), in combination
and in isolation, influence STT of ozone over western North America. Strictly speaking, warming SSTs in high emission
scenarios such as RCP8.5 result from the increased GHG emissions. However, when considered independently of each other,
the SSTs and the GHGs have distinct impacts on the future atmosphere, with the SSTs being disproportionately responsible
for future subtropical jet changes and amplification of the BDC's shallow branch (Oberländer et al. 2013; Chrysanthou et al.
2020) and the GHGs being primarily responsible for production of stratospheric ozone and amplification of the BDC's deep
branch (Winterstein et al. 2019; Abalos et al. 2021; Dietmüller et al. 2021). Therefore, as is shown in Section 3, each forcing,
either dynamically or chemically, influences processes that affect STT over western North America. Section 4 synthesizes the
results, namely that the RCP8.5 GHGs are primarily responsible for future increases in lower tropospheric O3S over western
North America despite the RCP8.5 SSTs disproportionately accounting for future dynamical changes in the troposphere and
stratosphere, including those associated with the North Pacific jet's spring transition.
**2 Methods**
We compare output from three different 60-year integrations using NCAR WACCM (Table 1). The version of WACCM used
in this study uses a horizontal resolution of 1.9° latitude by 2.5° longitude with 70 vertical layers and a model top near 140 km
(Mills et al. 2017, Richter et al. 2017). These experiments do not include an internally generated or prescribed Quasi-Biennial
Oscillation; the climatological tropical stratospheric winds are weakly easterly. WACCM has fully interactive chemistry in the
middle atmosphere using the Model for Ozone And Related chemical Tracers (MOZART3) and a limited representation of
tropospheric chemistry (Kinnison et al. 2007). The chemistry module in WACCM includes a stratospheric ozone tracer (O3S),
which is used to quantify STT of ozone. O3S is set equal to the fully interactive stratospheric ozone at each model timestep.
Once it crosses the tropopause, O3S decays at a tropospheric chemistry rate and is lost due to dry deposition.

To isolate the signal of atmospheric tracers to external forcings above the 'noise' of internal atmospheric variability, we have
run "time-slice" simulations forced by fixed SSTs, allowing us to both generate longer simulations than more computationally
expensive coupled atmosphere-ocean simulations, and to remove the interannual variability driven by the ocean (e.g.,



variability due to El Niño Southern Oscillation, ENSO). Each time-slice simulation has been run for 60 years, with 10 years of spin-up (which is sufficient for initialized atmosphere-only runs).

| Name | Experiment type | SST years | GHG year | Methane (ppb) | Nitrous oxide (ppb) | Carbon dioxide (ppm) | $Cl_y$ (ppb) |
|------|-----------------|-----------|----------|---------------|---------------------|----------------------|--------------|
| EXP1 | Preindustrial | 1840-1870 | 1850 | 790 | 275 | 285 | 0.46 |
| EXP2 | RCP8.5 | 2070-2090 | 2090 | 3632 | 421 | 844 | 1.36 |
| EXP3 | RCP8.5 SSTs | 2070-2090 | 1850 | 790 | 275 | 285 | 0.46 |

Table1: Each experiment is prescribed with fixed repeating annual cycles of the time averaged SST from the years listed in column three. Greenhouse gas mixing ratios coinciding with the years indicated in column four are shown for four of the gases in columns five through eight.

The first experiment (EXP1) is a preindustrial control simulation forced with year 1850 GHGs and a fixed repeating annual cycle of SSTs and sea ice created from the time averaged 1840 to 1870 period. The second experiment (EXP2) is forced with a fixed repeating annual cycle of SSTs/sea ice based on the time averaged 2070 to 2090 period from a fully-coupled run of the same version of WACCM, and GHG concentrations at year 2090 from the RCP8.5. The RCP8.5 scenario represents a "worst-case" future scenario in which the radiative forcing imbalance between year 2100 and 1850 is 8.5 W m$^{-2}$ due to marked increases in concentrations of carbon dioxide, nitrous oxide, and methane by the end of the century (Van Vuuren et al. 2011). We chose this extreme scenario in order to simulate the "upper bounds" of the response. There are also increased concentrations of ozone-depleting substances (ODS; e.g., chlorofluorocarbons) relative to the preindustrial experiment, due to the long lifetimes of these substances, which were emitted prior to the Montreal Protocol. Non-methane ozone precursor emissions, the solar flux, and stratospheric aerosol concentrations are held fixed to year 1850 levels. The difference between EXP2 and EXP1 includes the atmospheric response to future GHGs and future SSTs.

One additional experiment is used to disentangle the atmospheric response to future GHGs and future SSTs. This third experiment (EXP3) is identical to the RCP8.5 experiment (EXP2), except that GHGs are held fixed to year 1850 concentrations, meaning the RCP8.5 SST perturbation is the only forcing. By comparing EXP3 to EXP1, we can isolate the atmospheric response to future SSTs only. This response is referred to as "RCP8.5 SSTs" throughout this study. If we instead compare the experiment in which the RCP8.5 SSTs are the only forcing (EXP3) to the full RCP8.5 experiment (EXP2), we approximate the atmospheric response to future GHGs. This response is referred to as "RCP8.5 GHGs" throughout this study. Note that if one wanted to assess the linearity of the RCP8.5 SST and RCP8.5 GHG responses to the full RCP8.5 response, another experiment would be required, one identical to the RCP8.5 experiment, but with the SSTs held fixed to the time averaged 1840-1870 state used in the preindustrial control, EXP1. In the absence of this experiment, our "RCP8.5 GHG responses" should be thought of as approximations. Given that the RCP8.5 SST only experiment (EXP3) is used as the





reference case to derive the atmospheric response to RCP8.5 SSTs (by comparison to the preindustrial control, EXP1) and as
the reference case to approximate the atmospheric response to RCP8.5 GHGs (by comparison to the full RCP8.5 experiment,
EXP2), by construction, the RCP8.5 SST responses and the RCP8.5 GHG responses are perfectly additive; these response
together recover the full RCP8.5 response.

**2.1 Decomposing the jet into late winter, spring, and summer phases**
Breeden et al. (2021) showed that the mass of stratospheric air entering the lower troposphere over western North America is
three times larger during the jet's spring transition phase as opposed to its late winter or summer phases. This peak in mass
transport is associated with enhanced synoptic scale wave activity in the upper troposphere, tropopause folds that reach deeper
into the troposphere, and a deeper planetary boundary layer. Because the seasonal evolution of the North Pacific jet impacts
STT over western North America, in all of our analyses, we consider changes in all fields as a function of the three phases of
the seasonal transition of the North Pacific jet as they are defined in Breeden et al. (2021). Therefore, the differences in transport
arising from timing of the jet transition are inherently taken into account.

Figure 1 shows the seasonal evolution of the North Pacific jet in the preindustrial control and in the RCP8.5 experiment. The
jet is separated into winter, spring, and summer phases using the principal component time series associated with the first
empirical orthogonal function (EOF) of the daily 200 hPa zonal winds averaged over the North Pacific region (100°E - 280°E
and 10°N - 70°N). The zonal wind anomalies used for the EOF analysis are calculated with respect to the February to June
years 11-60 average, rather than a daily climatology, in order to deliberately preserve the seasonal cycle that emerges as the
first EOF. The associated principal component time series (PC1), calculated by projecting the gridded zonal wind for either
the preindustrial control (Fig 1d) or the RCP8.5 experiment (Fig 1e) at each time step onto each experiment's EOF1, are
smoothed with a five-day running mean.



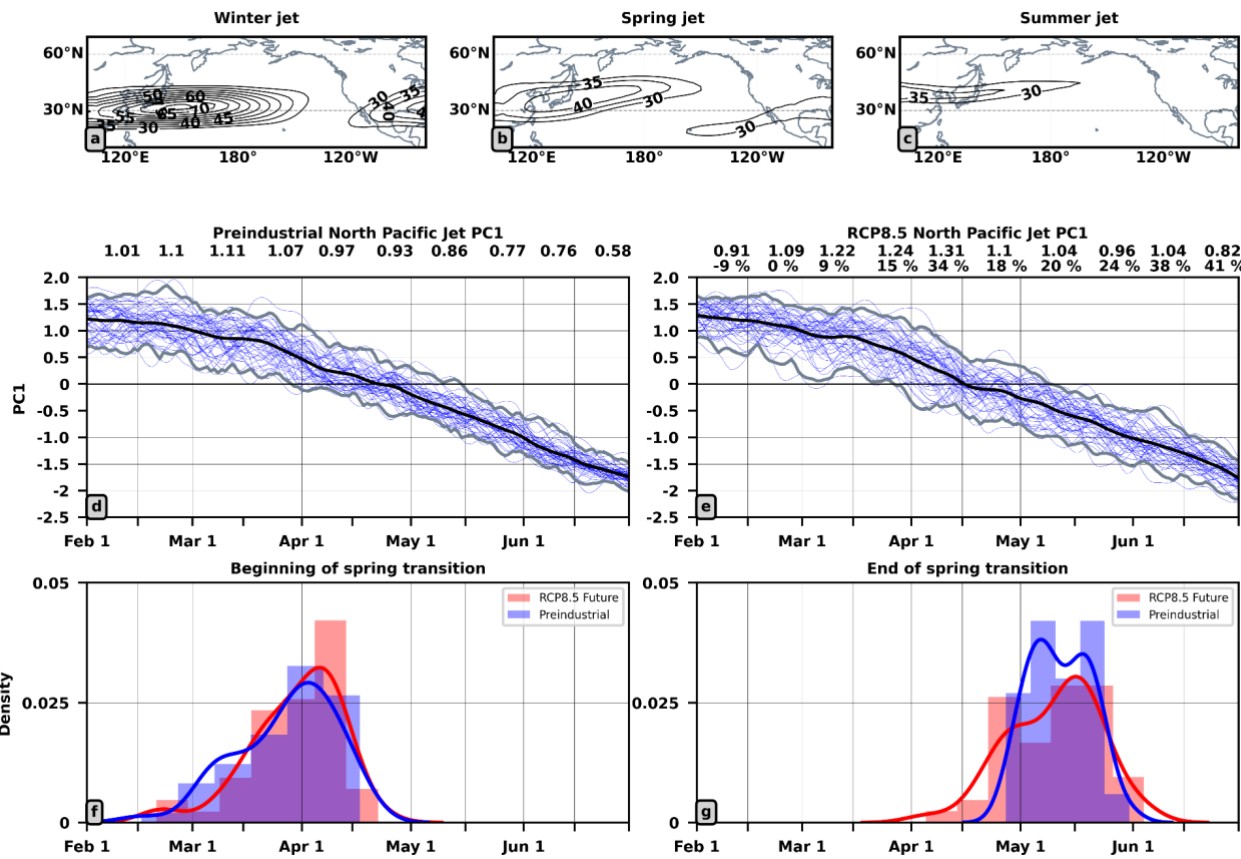

**Figure 1:** Spring transition of the North Pacific jet in the preindustrial control (EXP1) and the RCP 8.5 experiment (EXP2). (a-c) shows preindustrial 200 hPa zonal winds subsampled for the jet's winter phase (PC1 > 1σ), spring phase (PC1 < 0.5σ and > -0.5σ), and summer phase (PC1 < -1σ). (d) shows the temporal evolution of PC1 in the preindustrial control with the mean PC1 shown in black, PC1 for each year shown in blue, and the 2.5% and 97.5% confidence intervals calculated by bootstrapping with replacement (10,000 times for each day) shown in gray. The average difference between the 2.5% and 97.5% confidence intervals for each ~ two week period (referred to as "spread") are shown above panel (d). Panel (e) is the same as panel (d), but for RCP8.5. In addition, the percent change between the RCP8.5 and preindustrial "spread" is also printed above panel (e). Panel (f) and panel (g) are kernel density plots estimating when the spring transition begins (PC1 = 0.5σ) and when the spring transition ends (PC1 = -0.5σ), respectively.

The winter jet is present when PC1 > 1 standard deviation (σ), during which the Pacific jet is strong and narrow (Fig. 1a). The spring jet is present when PC1 < 0.5 σ and > -0.5 σ, at which point the subtropical jet weakens and shifts north, and the secondary subtropical jet maximum extends between Hawaii and Western North America (Fig. 1b). The summer jet is present when PC1 < -1 σ (Fig. 1c). The jet weakens substantially and remains shifted poleward, and the secondary jet maximum over North America weakens. The structure of winter, spring, and summer jets (Figs. 1a-c) compares well with that from 1958-2017 Japanese Reanalysis-55 data (cf. Fig. 2, Breeden et al. 2021) as does the timing of the phase changes (Figs 1d-g, cf. Figs. 1 and 3, Breeden et al. 2021).




The RCP8.5 North Pacific jet exhibits statistically significant (Fig. A1) increases in variability compared to the preindustrial
control during much of spring and summer (Fig. 1e). Recomputing Figure 1e using the EXP3, which includes RCP8.5 SSTs,
but preindustrial GHGs, confirms that the changing jet variability is associated with the RCP8.5 SSTs (not shown). Despite
these changes in variability, there is no statistically significant change in when the spring transition begins (Fig. 1f) or ends
(Fig. 1g, Fig. A1). The median start date for the preindustrial control is March 31$^{st}$ with a σ of +/-13 days and the median end
date is May 11$^{th}$ +/- 8 days. For RCP8.5, the median start date is April 1$^{st}$ with a σ of +/- 12 days and the median end date is
May 13$^{th}$ +/- 13 days. Consistent with Fig. 1g, the enhanced jet variability due to RCP8.5 conditions manifests as a broader
distribution of median end dates. With no robust change in the timing of the spring transition, the calendar dates corresponding
to the late winter, spring, and summer jet phases are similar amongst the experiments. Therefore, in all subsequent figures,
anomalies are calculated by binning each individual experiment's data according to that experiment's late winter, spring, and
summer days, time averaging the data within each bin, and then differencing between the jet phase (e.g., late winter) bins from
two different experiments (e.g., EXP2 minus EXP1). This approach would not be possible if, for instance, the annually
averaged late winter end date from EXP2 was 10 days after that from the EXP1. Similar results to those shown in figures 2-6
can be obtained by comparing like months (e.g., February-March, April-May) from two different experiments (not shown).
However, we choose to show our results according to jet phase so that the STT inherently associated with each phase is
accounted for.

Note that while no changes in the timing of the spring transition are found in these simulations, spring transition timing is
heavily influenced by ENSO (Breeden et al. 2021) and neither ENSO variability nor its response to climate change are included
in these experiments. Hence, Figure 1 does not provide a comprehensive answer to the question of how RCP8.5 forcing
modifies the timing of the spring transition, which will have to be assessed in subsequent research.

**2.3 Residual advection, two-way isentropic mixing, production and loss of O3S**
To quantify the contributions of the residual advection, two-way isentropic mixing, and production and loss to the total O3S
response, we calculate the terms in the Transformed Eulerian Mean (TEM) continuity equation for zonal mean tracer transport
given by Andrews et al. (1987, equation 9.4.13) and discussed by Abalos et al. (2013). Daily data, time averaged from the 6-
hourly fields, is used to calculate each term. These terms are shown in Eq. (1):
$\frac{\partial \bar{\chi}}{\partial t} + \overline{v^*} \frac{\partial \bar{\chi}}{\partial y} + \overline{w^*} \frac{\partial \bar{\chi}}{\partial z} = P - L + e^{-z/H} \nabla \cdot M$ ,                    (1)
where overbars denote zonal averages, $\chi$ denotes the ozone concentration in parts per billion, $P$ denotes chemical production
and $L$ chemical loss, $H$ is the scale height equal to 7 kilometers, $y$ and $x$ are the meridional and zonal cartesian coordinates, $z$



is log-pressure height, $\nabla$ is the divergence operator, and $M$ is the two-way isentropic mixing vector with meridional and vertical
components given by Eq. (2) and (3):
$$\frac{\partial \bar{M}}{\partial y} = -e^{-\frac{z}{H}}(\overline{v'\chi'} - \frac{\overline{v'T'}}{S}\frac{\partial \bar{\chi}}{\partial z})$$    (2)
$$\frac{\partial \bar{M}}{\partial z} = -e^{-\frac{z}{H}}(\overline{w'\chi'} + \frac{\overline{v'T'}}{S}\frac{\partial \bar{\chi}}{\partial y})$$    (3)
where primes denote deviations from the zonal average, $v$ and $w$ are the meridional and vertical velocities, $S$ equals $(H \cdot$
$N^2)/R$ in which $N^2$ is the Brunt-Väisälä frequency and $R$ is the gas constant equal to 287 m$^2$/s$^2$/K. The residual circulation
velocities $(\underline{v}^*, w^*)$ are given by Eq. (4) and (5):
$$\overline{v^*} = \bar{v} - \frac{1}{\rho_0}\frac{\partial}{\partial z}(\frac{\rho_0 \overline{v'\theta'}}{\partial \theta / \partial z})$$    (4)
$$\overline{w^*} = \bar{w} + \frac{1}{a\cos\varphi}\frac{\partial}{\partial \varphi}(\frac{\cos\varphi \overline{v'\theta'}}{\partial \theta / \partial z})$$    (5)
where $\rho_0$ is log-pressure density and $\theta$ is potential temperature and $a$ is Earth's radius.

**3 Results**
**3.1 Lower tropospheric O3S responses**
To better understand how climate change may influence the amount of stratospheric ozone making it into the lower free
troposphere over western North America, Figure 2 shows the 700 hPa O3S responses to RCP8.5 forcing, RCP8.5 SSTs, and
RCP8.5 GHGs for the late winter, spring, and summer North Pacific jet phases. In the preindustrial control climatology, lower
tropospheric O3S increases from low to high latitudes regardless of season, and mixing ratios are largest over western North
Pacific during the jet's spring phase, mimicking the observed seasonal maximum in deep STT over this region (Fig 2 black
lines; Skerlak et al. 2014; Breeden et al. 2021).

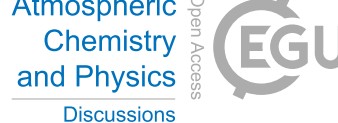

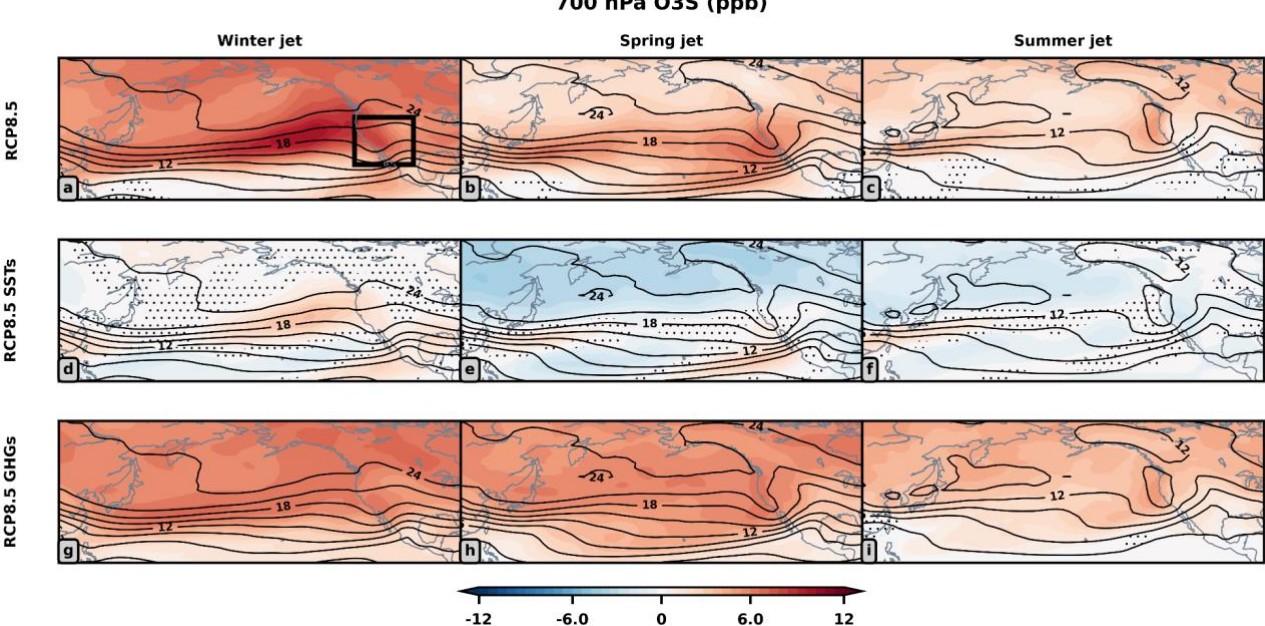


Figure 2: 700 hPa O3S (ppb) response to RCP8.5 boundary conditions shown in shading. (a-c) show the response to RCP8.5 conditions, (d-f) response to
RCP8.5 SSTs, and (g-i) response to RCP8.5 GHGs. The 700 hPa O3S preindustrial control seasonal climatologies are overlaid in black. Non-stippled grid
points are statistically significant at a 5% significance threshold using a bootstrapping hypothesis test (Efron and Tibshirani 1994) in which the two samples
being compared are resampled 1,000 times at each grid point. The phases of the jet are shown in successive columns.

RCP8.5 forcing increases lower tropospheric O3S over most of the longitudinal domain shown and over much of the
hemisphere (not shown) during all three seasons (Fig. 2a-c). The RCP8.5 response is  largest in late winter, during which there
is up to a 50% increase in O3S over the North Pacific and a 39% increase over western North America (25N-45N, 235E-260E,
Fig. 2a box). Although the responses are  smaller in absolute magnitude during spring and summer compared to winter, they
still coincide with roughly 10-35% change relative to climatology. Notably in spring, the largest increases are centered over
western North America (Fig. 2b).

The RCP8.5 SSTs (Fig. 2d-f) increase O3S by approximately 15% over the eastern North Pacific during the jet's late winter
phase (Fig. 2d), explaining a portion of the aforementioned 50% increase in late winter RCP8.5 O3S over this region (c.f. Fig.
2a). Over the low latitude eastern North Pacific, close to Baja California/Mexico, the RCP8.5 SSTs promote large increases in
O3S during the jet's winter and spring phases relative to preindustrial climate (Fig. 2d-e). Conversely at high latitudes, O3S
does not change during the late winter phase in response to RCP8.5 SSTs, and decreases by roughly 10% during both the jet's
spring and summer phases. In summary, the RCP8.5 SSTs can explain a portion of the full RCP8.5 response, but clearly not
the bulk of it.





The response to RCP8.5 GHGs alone accounts for the majority of the full RCP8.5 700 hPa O3S response (Fig. 2g-i). Larger
O3S increases develop during the jet's late winter and spring phases compared to summer. Recall that the full RCP8.5 response
is a linear combination of the RCP8.5 SST (Fig. 2g-i) and RCP8.5 GHG contributions (Fig. 2j-l). Both SSTs and GHGs
increase O3S over the central North Pacific and western North America during the jet's late winter phase, but have competing
effects on O3S during the jet's spring and summer phases. To better understand the future changes in free tropospheric O3S
and the relative roles of SST and GHG changes, the next sections consider in more detail how the North Pacific jet and the
lower stratospheric ozone reservoir respond to climate change.

**3.1 Changes in the upper troposphere and lower stratosphere**
RCP8.5 conditions accelerate, narrow, and elongate the late winter North Pacific jet towards western North America at 200
hPa (Fig. 3a). This change is robust to varying severities of climate change (RCP4.5, Harvey et al. 2020; RCP6.0, Akritidis et
al. 2019; and RCP8.5, Matsumura et al. 2021). Contrary to what takes place during the late winter period, the subtropical jet
shifts equatorward during the jet's spring and summer phases (Fig. 3b-c). At lower latitudes, westerly anomalies form over the
subtropical eastern Pacific/central America, where there is a climatological minima in the 200 hPa zonal wind (Fig. 3a-c). This
response is present during all three jet phases  and strengthens from late winter through summer (Fig. 3a-c).

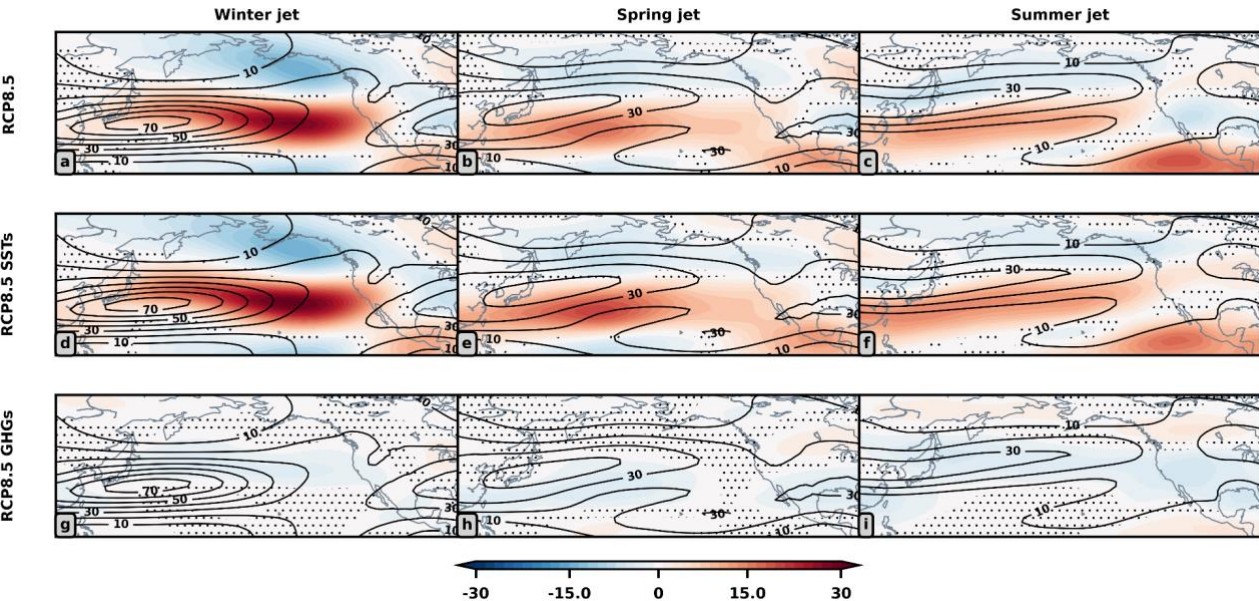

Figure 3: As in Figure 2, but for the 200 hPa zonal winds.



The total RCP8.5 200 hPa zonal wind response is dominated by the contribution from the RCP8.5 SSTs (Figs. 3d-f) with the
GHGs (Figs. 3g-i) playing a comparatively minor role. The strong influence of the future SSTs on the wind response arises in
part because the RCP8.5 SST forcing is associated with almost all of the ~9-11 Kelvin warming of the tropical upper
troposphere and the amplified Arctic surface warming (Fig. S1), and so dominates the influence on meridional temperature
gradients and associated circulation changes that drive heat transport. Another consideration is that the zonal asymmetries in
the pattern of RCP8.5 SSTs prescribed in the experiments, particularly those over the tropical Pacific resembling El Niño (Fig.
S2), elicit teleconnections (e.g., Pacific North America (PNA) teleconnection pattern) that modify the upper tropospheric
circulation. The impact of the GHGs alone on the 200 hPa winds is small, although the RCP8.5 GHGs do have a large
(compared to climatology) effect on the zonal wind over western North America during the jet's summer phase (Fig. 3i),
illustrating that purely chemical changes in the atmosphere are capable of having significant dynamical impacts.

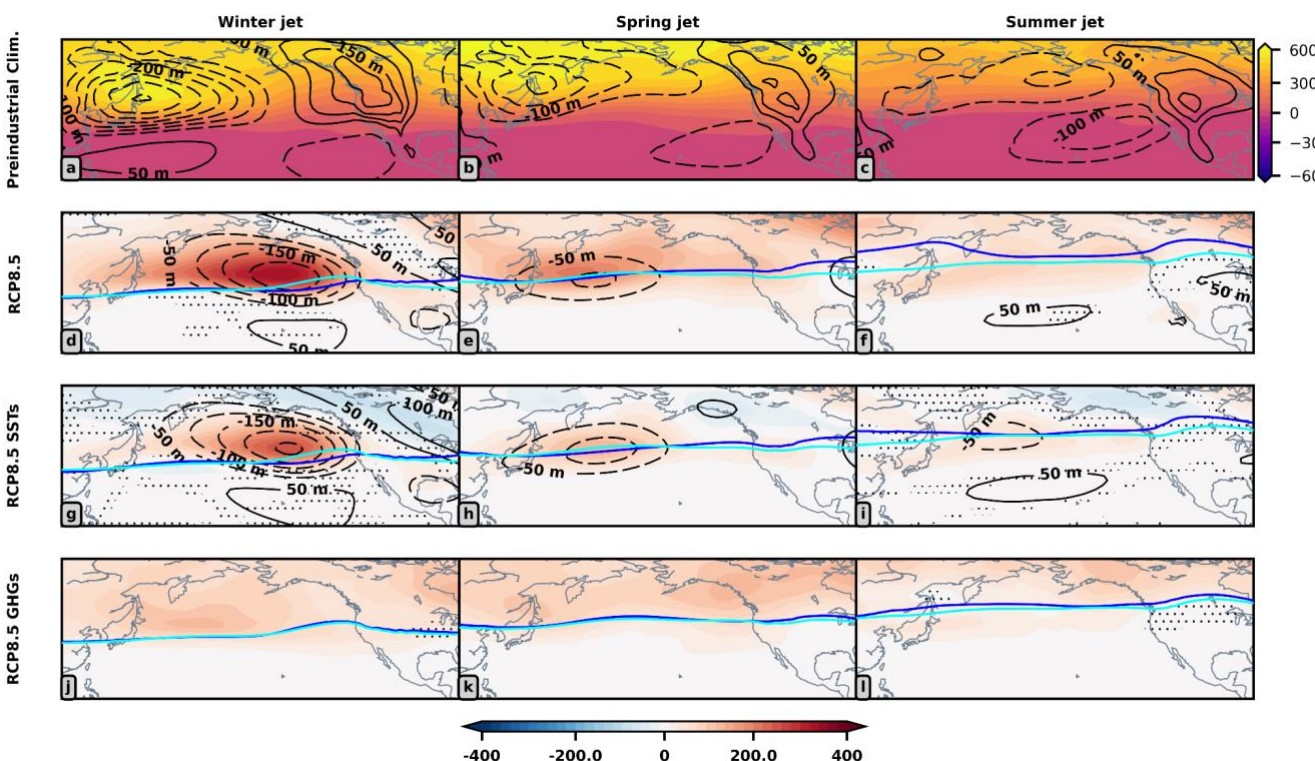


Figure 4: 200 hPa O3S (shaded) and stationary wave (contous, long-term zonal mean geopotential height removed, GEOPx) responses to RCP8.5 boundary
conditions. (a-c) show the preindustrial climatologies of O3S in alternate shading and the climatological stationary wave in contours (d-l). (d-f) show O3S
response to RCP8.5 conditions in shading and stationary wave anomalies are contoured, (g-i) same, but for RCP8.5 SSTs, and (j-l) same, but for RCP8.5
GHGs. Non-stippled grid points are statistically significant O3S anomalies at a 5% significance threshold using a bootstrapping hypothesis test. The phases
of the jet are shown in successive columns. The preindustrial control thermal tropopauses for each season are shown in blue and anomalous tropopauses are
shown in cyan.






Figure 4 shows how RCP8.5 conditions modify 200 hPa O3S, allowing us to see both tropospheric and stratospheric ozone
changes; at 200 hPa, the stratosphere is poleward of the anomalous thermal tropopause (cyan lines), which can be compared
with the preindustrial  thermal tropopause (blue lines) in each season. 200 hPa O3S equatorward of the tropopause has already
been transported into the troposphere  and can be lost due to dry deposition and chemical loss or transported back to the
stratosphere by reversible mixing processes.

In the preindustrial control, EXP1, O3S maxima and minima are co-located with the troughs and ridges of the climatological
stationary wave (Figs. 4a-c). This is particularly clear in late winter, during which O3S mixing ratios exceed 600 ppb over the
wave-1 scale trough of the climatological stationary wave, the Aleutian Low (Fig. 4a). O3S mixing ratios are, on the other
hand, reduced over the climatological Alaskan Ridge. Slightly out of view in Fig 4a is a climatological wave-2 scale trough
that resides over the Baffin Bay and Greenland; an O3S maxima is found over this region as well (Fig. 4a).  As suggested by
Reed (1950; see also Schoeberl and Kreuger 1983 and Salby and Callaghan 1993), horizontal advection and vertical motion
associated with waves act to concentrate ozone in troughs and reduce it over ridges. The climatological stationary wave
influences the 200 hPa composition of O3S in this way.

RCP8.5 conditions increase lower stratospheric O3S over much of the hemisphere during all seasons (Fig. 4d-f). The largest
regional increase is a doubling of O3S over the North Pacific during the jet's late winter phase (Fig. 4a, 4d). This regional O3S
increase is co-located with the trough of an anomalous tropical-extratropical planetary-scale wave, whose signature is apparent
in the zonal wind response (Fig. 3) and the stationary wave response (Fig 4, black contours). As the amplitude of this wave
diminishes during the spring and summer phases, so does the lower stratospheric O3S maxima (Fig. 4e-f). The RCP8.5 O3S
response is mostly contained in the lower stratospheric (i.e., poleward of the tropopause) trough during the jet's late winter
phase, but in the absence of strong meridional potential vorticity gradients such as the high-latitude polar stratospheric
westerlies (Manney et al. 1994; Salby and Callaghan 2007) or the subtropical jet stream (Bönisch et al. 2009), which serve as
transport barriers, the O3S response "smears out" during spring and summer, becoming more evenly distributed around the
200 hPa thermal tropopause (Fig. 4e-f).

The RCP8.5 SSTs are almost solely responsible for the development of the anomalous planetary wave and are therefore a key
reason why there are zonal asymmetries in the lower stratospheric ozone reservoir (Fig.4g-i). Similar effects of large-scale
planetary wave trains on lower stratospheric ozone have been noted in relation to ENSO (Zhang et al. 2015; Albers et al. 2022).
The RCP8.5 SST forcing considered in this study displays SST warming globally, but contains some zonal asymmetries, one
of them being an El Niño-like eastern tropical Pacific warming (Fig. S2). This zonal asymmetry may explain why the planetary
wave response to the RCP8.5 SSTs during late winter (Fig. 4g) resembles the PNA wave train known to develop with El Niño
(albeit the Canadian ridge in Fig. 4g is displaced east relative to PNA Canadian ridge). Note though that there is large inter-



model and inter-generational (CMIP5 vs. CMIP6) spread in how ENSO responds to climate change (Beobide-Arsuga et al.
2021; Cai et al. 2022), suggesting that this planetary wave response could vary amongst climate models, should it in fact be
related to the El Niño-like warming superimposed on the global SST increase (Fig. S2).

One interesting quality of the planetary wave forced by the RCP8.5 SSTs is that it constructively interferes with the
climatological stationary wave, promoting additional upward planetary wave propagation into the stratosphere. The coherence
between the forced (by the RCP8.5 SSTs) trough over the North Pacific and the climatological Aleutian Low as well as the
coherence between the forced ridge and the climatological Alaskan Ridge (Fig. 4k) promote the anomalous upward planetary
wave propagation. This wave tilts westward with increasing height at high-latitudes (not shown), particularly in the middle to
upper stratosphere, suggesting not only that anomalous upward propagation is taking place, but that the forced wave is
modifying the stratospheric mean flow, thereby reinforcing the residual mean meridional circulation associated with the BDC.
In conjunction with gravity waves, changes to the planetary scale stationary wave are expected to enhance the BDC
(Oberländer et al. 2013). Moreover, Chrysanthou et al. (2020) showed that the RCP8.5 SSTs account for nearly half of the
acceleration of the BDC's residual mass streamfunction above 30 hPa between 0°N and 60°N. Such a change to the BDC's
deep branch likely requires a perturbation to the longwaves, zonal wavenumbers-1 and 2, which Fig. 4k shows is occurring.

Contrary to the RCP8.5 SSTs, the RCP8.5 GHGs have little effect on the planetary-scale eddies and elicit more zonally
symmetric O3S responses (Fig. 4j-l). The lower stratospheric O3S response to the GHGs develops in part due to net chemical
production of stratospheric ozone, likely associated with the large RCP8.5 methane increase, which enhances O3 mixing ratios
in the extratropical stratosphere (Morgenstern et al. 2018), and changes in transport associated with the BDC's deep branch.






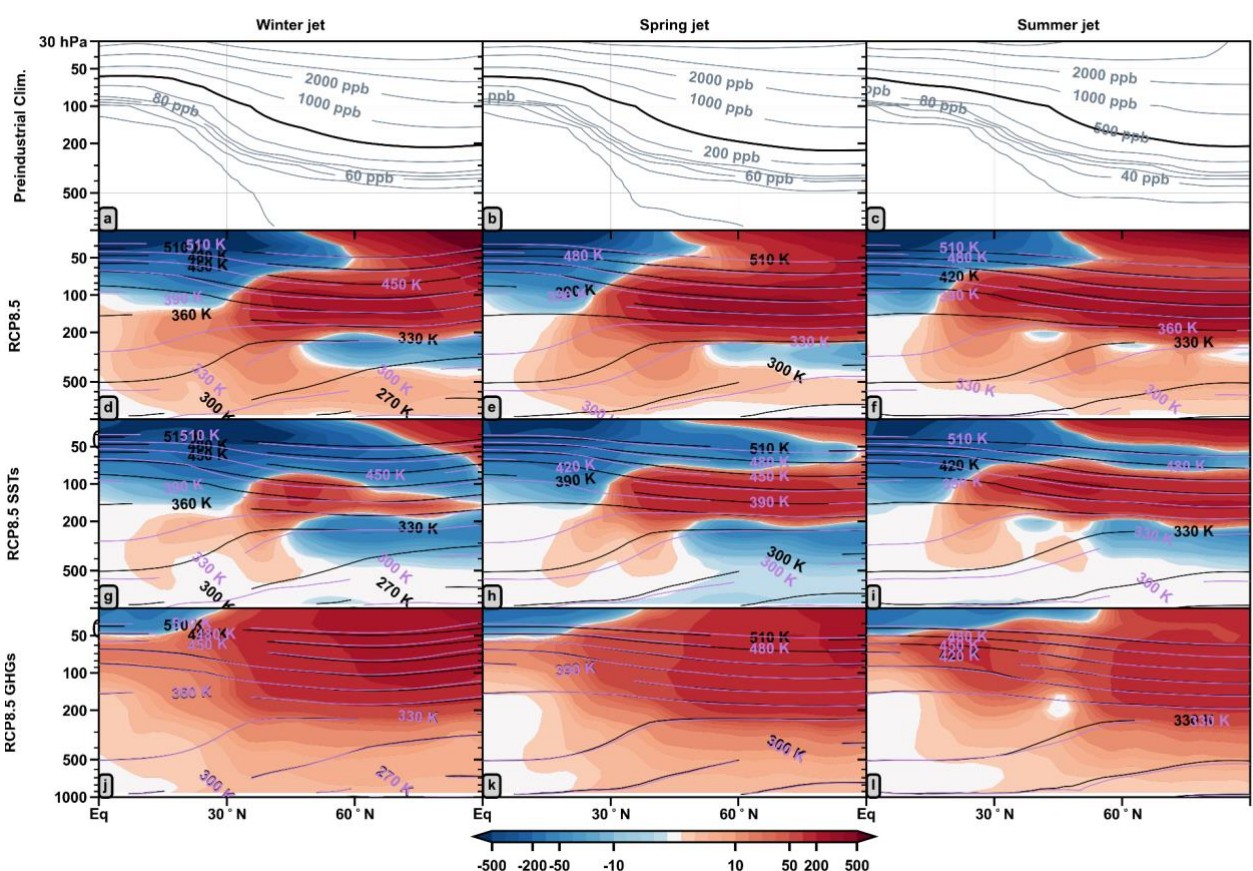

Figure 5: Transects of the O3S anomalies and isentropes averaged between 235°E and 260°E (over western North America). (a-c) show preindustrial climatologies of O3S; contour intervals are 20, 40, 60, 80, 100, 200, 500 (shown in thick black contour), 1000, 2000, 3000, and 4000 ppb. (d-f) show O3S response to RCP8.5 forcing in shading, with preindustrial isentropes shown in black, and the anomalous isentropes in magenta, (g-i) show the same, but for RCP8.5 SSTs, and (j-l) same, but for RCP8.5 GHGs. Non-stippled grid points are statistically significant O3S responses at a 5% significance threshold using a bootstrapping hypothesis test. The phases of the jet are shown in successive columns.

To further clarify how the lower stratospheric reservoir responds to RCP8.5 conditions, Figure 5 shows latitude-pressure transects of O3S anomalies and isentropes averaged between 235°E and 260°E (over western North America). Climatologically, extratropical lower stratospheric O3S mixing ratios are larger during winter and spring (Fig. 5b), following from transport by the BDC's deep branch (Ray et al. 1999; Hegglin and Shepherd 2007; Bönisch et al. 2009; Butchart 2014; Konopka et al. 2015, Ploeger and Birner 2016; Albers et al. 2018). During summer in climatology, enhanced isentropic mixing between the tropical and extratropical lowermost stratosphere (Hegglin and Shepherd 2007; Abalos et al. 2013) and rising tropopause heights (Schoeberl et al. 2004) act to flush ozone out of the lowermost stratosphere.





During every jet phase, RCP8.5 conditions reduce O3S in the low latitude stratosphere while promoting accumulation of O3S
at high latitudes (Fig. 5d-f). Some of this O3S accumulating in the extratropical lower stratosphere may enter the troposphere
along the subtropical upper tropospheric/lower stratospheric isentropes (e.g., 360 K). Both the GHGs and SSTs play a role in
making this happen. The upper tropospheric warming induced by the RCP8.5 SSTs depresses the isentropes (e.g., 360 K) to
lower altitudes, enhancing the access of the troposphere to lower stratospheric air (Fig. 5g-i), where wave breaking is able to
transport the ozone into the subtropical and tropical upper troposphere (e.g., Waugh and Polvani 2000, Albers et al. 2016 and
references therein). The GHGs on the other hand mainly contribute by more broadly enhancing the extratropical lower
stratospheric O3S concentrations (Fig. 5j-l).
O3S is reduced near the extratropical tropopause in all seasons in response to RCP8.5 forcing (Figs. 5d-f). This is associated
with the increased height of the tropopause (Abalos et al. 2017) resulting  from the RCP8.5 SSTs. Due to steep vertical
gradients in tracers near the tropopause (e.g., Pan et al. 2004), taking the difference between an experiment with a lifted
tropopause (EXP2 or EXP3) and an experiment without this feature (preindustrial control, EXP1) amounts to taking the
difference between relatively O3S depleted tropospheric air and O3S rich stratospheric air, hence the negative O3S anomalies
develop near the tropopause (Figs. 5d-i). This negative O3S response can largely be removed by remapping the vertical axis
of each data field used to make, for instance, Figs. 5d-f (zonally averaged RCP8.5 O3S and preindustrial O3S) to tropopause-
relative coordinates (meters above or below the thermal tropopause), then taking the difference between these two modified
data fields, and remapping this set of anomalies (axes: tropopause-relative x latitude) to a log-pressure coordinate system (axes:
pressure x latitude) (Abalos et al. 2017). Using annual cycles of thermal tropopause and O3S data, which should help to smooth
out the large hourly/daily fluctuations in these fields near the tropopause, the aforementioned procedure was applied to a
zonally averaged transect over the North Pacific (Fig S3) and applied at all grid points at 200 hPa (Fig. S4) and 300 hPa (Fig.
S5). While this tropopause-relative analysis does remove the majority of the negative O3S response associated with the
tropopause lift, the strong O3S zonal asymmetries associated with the planetary wave response to the RCP8.5 SSTs persist,
namely the negative O3S response corresponding to the planetary wave's ridge near Alaska (cf. Fig. 4k, Fig. S5e). This analysis
corroborates that the higher tropopause in RCP8.5 is largely responsible for the presence of the negative O3S response in the
extratropical upper troposphere/lower stratosphere, however not entirely, as we find that a portion of this negative O3S is
associated the anomalous planetary wave's zonally asymmetric effects on the upper tropospheric/lower stratospheric O3S
distribution.





## 3.3 Zonally symmetric changes

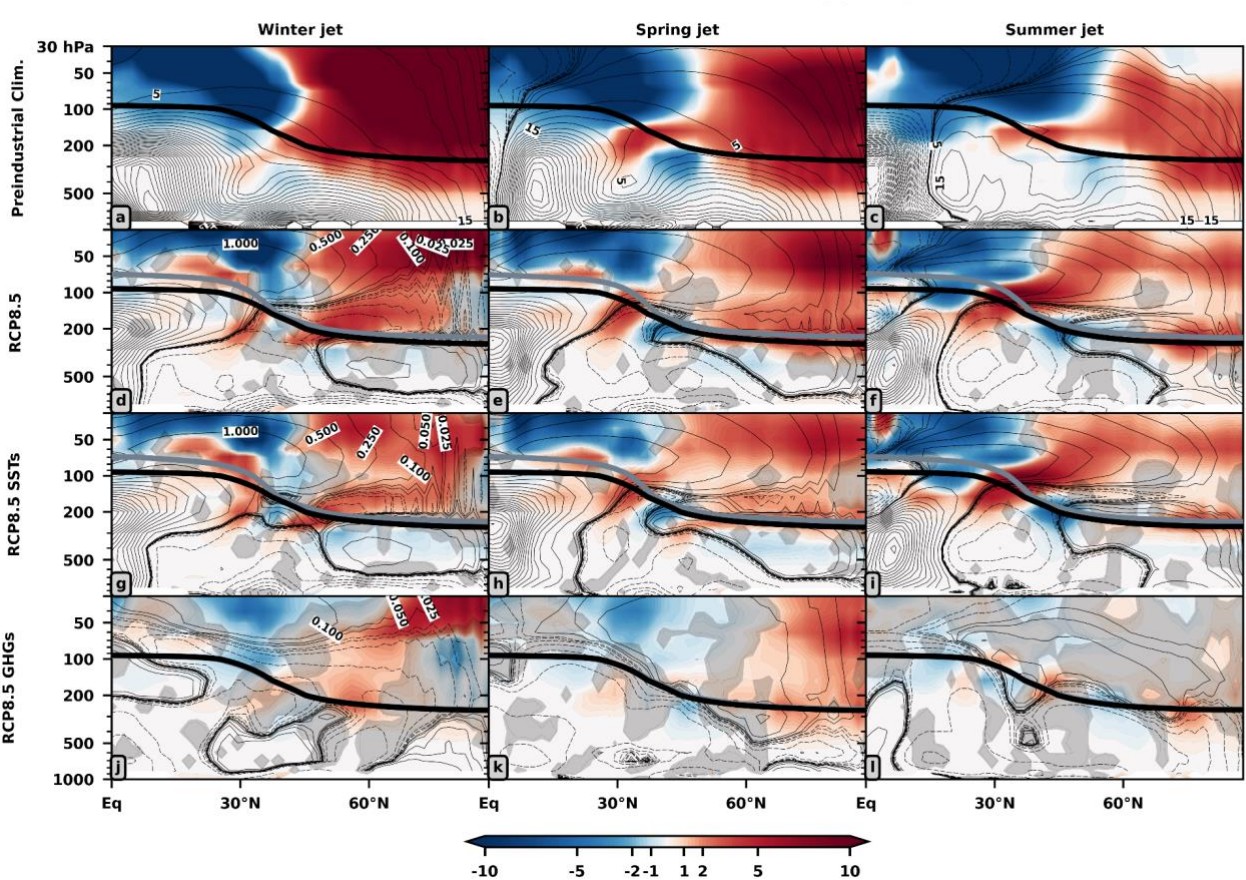

Figure 6: Residual advective O3S tendencies (shading) and residual mass streamfunction (contours). (a-c) show preindustrial residual advective O3S tendencies in shading with the climatological residual mass streamfunction overlaid in black contours. The color scale is the same for the climatology and anomalies. The contour intervals for the residual mass streamfunction in all panels are 0.025, 0.05, 0.1, 0.25, 0.5, 1, 5, 10, 15, 20, 25 … (10^9 kg/s). (d-f) show the O3S tendency and streamfunction anomalies to RCP8.5, (g-i) show the same, but for RCP8.5 SSTs, and (j-l) same, but for RCP8.5 GHGs. Non-gray shaded grid points show statistically significant O3S anomalies at a 5% significance threshold using a bootstrapping hypothesis test. The phases of the jet are shown in successive columns. For each phase of the jet, the preindustrial control thermal tropopause is black and the anomalous tropopause is gray. Note that an anomalous tropopause is hardly visible in response to RCP8.5 GHGs as the SSTs are the forcing that modifies the tropopause.

The seasonal variability of both tropical (Abalos et al. 2013) and extratropical (Albers et al. 2018) lower stratospheric ozone tendencies is heavily influenced by upwelling and downwelling associated with BDC's residual mean meridional circulation component. This circulation is made up of a shallow and a deep branch. Transport associated with the shallow branch proceeds more horizontally and the air masses enter the stratosphere closer to the subtropics whereas transport associated with the deep branch is more vertical and the air masses enter the stratosphere through the deep tropics and descend at high-latitudes (Birner and Bönisch 2011). To quantify the influence of RCP8.5 forcing on these physical processes, Figure 6 shows the residual



mass streamfunction response to RCP8.5 forcing in black contours and in shading the local changes in O3S tendencies as
a result of transport by the residual mean meridional circulation terms in the TEM continuity equation ($\overline{v^*}\frac{\partial\overline{\chi}}{\partial y} + \overline{w^*}\frac{\partial\overline{\chi}}{\partial z}$ = As in
reanalysis (cf. Rosenlof 1995), in the preindustrial control, the tropical upward mass flux peaks in amplitude during boreal
winter when the residual mass streamfunction is strongest (Fig. 6a). As the zonal momentum budget changes in each
hemisphere during spring and summer, the tropical upward mass flux shifts into the northern hemisphere and the residual mass
streamfunction weakens and shifts downward towards the troposphere (Fig 6b, c). The negative O3S tendencies in the tropical
lower stratosphere track the latitudinal shifting of the tropical upward mass flux over time. The positive O3S tendencies in the
extratropical lower stratosphere associated with poleward transport of stratospheric ozone from its tropical source region peak
in amplitude during winter when the BDC's deep branch is strongest and weaken thereafter.

RCP8.5 forcing strengthens the shallow branch of the BDC during all three seasons, reducing tropical stratospheric O3S
tendencies (Fig. 6d-f). The RCP8.5 SSTs (Fig. 6g-i) are primarily responsible for the acceleration of the residual mass
streamfunction in the subtropical lower stratosphere (50 hPa/30°N) when compared against the RCP8.5 GHGs (Figs. 6j-l),
consistent with Oberländer et al. (2013) and Chrysanthou et al. (2020). The upper component of the Hadley Circulation near
150 hPa and 15°N accelerates, as previously reported by Abalos et al. (2020). All models they studied included this response.
This feature acts cooperatively with the reinforced BDC shallow branch to increase O3S transport through the subtropical
tropopause into the upper troposphere (200 hPa and 30°N), with the largest increase occurring during summer in response to
the RCP8.5 SSTs (Fig. 6i). The RCP8.5 GHGs accelerate  the deep branch well above 30 hPa during winter (Fig. 6j), its high-
latitude downwelling increases lower stratospheric O3S during spring (Fig. 6k), and then disappears by summer (Fig. 6i).

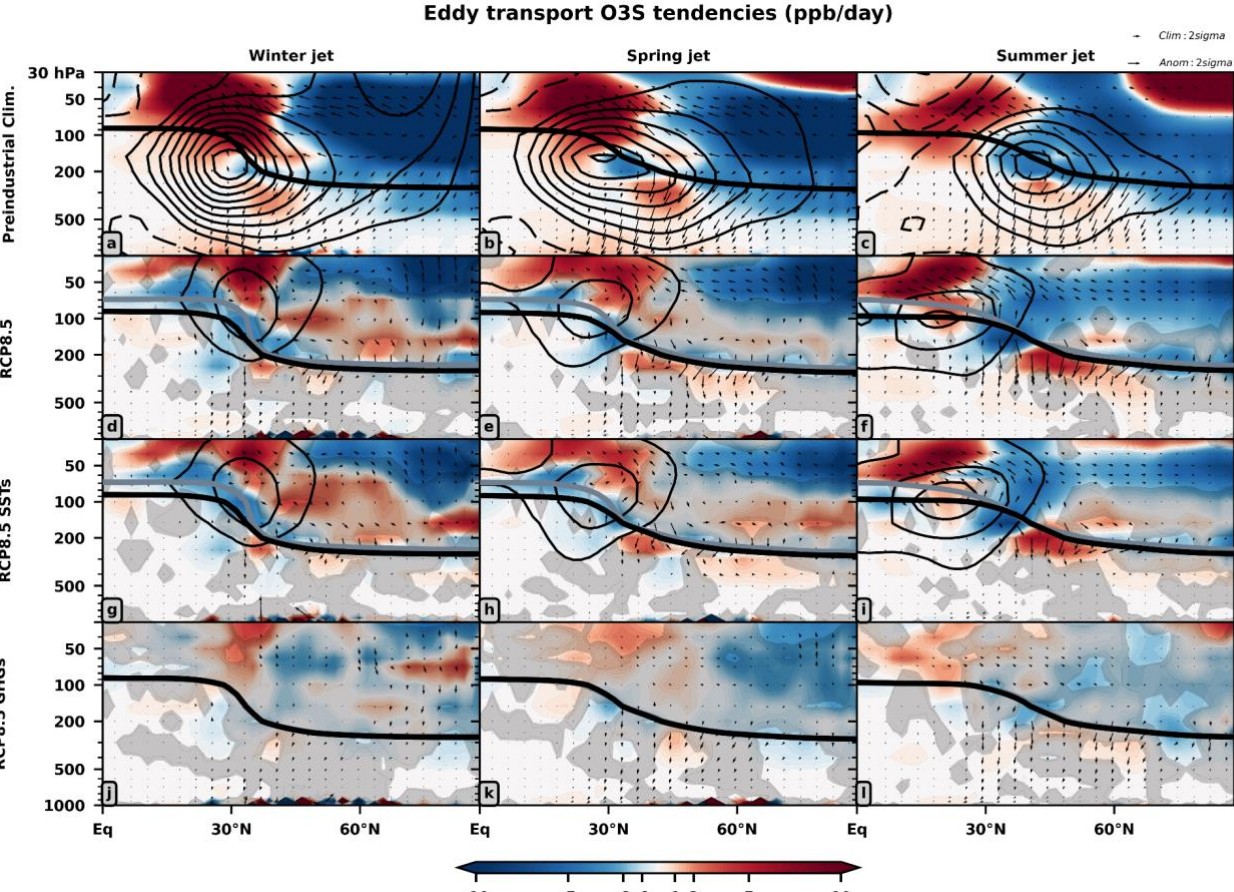

Figure 7: Two-way isentropic mixing O3S tendencies (shading) and zonal-mean zonal wind (contours). (a-c) show preindustrial O3S tendencies in shading with the climatological zonal wind overlaid in black (+/-5 m/s) and the components of the two-way isentropic mixing (-My,-Mz) shown as vectors. The color scale is the same for the climatology and anomalies. (d-f) show the O3S tendency and zonal-wind anomalies to RCP8.5, (g-i) show the same, but for RCP8.5 SSTs, and (j-l) same, but for RCP8.5 GHGs. Non-gray shaded grid points show statistically significant O3S anomalies at a 5% significance threshold using a bootstrapping hypothesis test.. The phases of the jet are shown in successive columns. For each phase of the jet, the preindustrial control thermal tropopause is black and the anomalous tropopause is gray.

Another aspect of the BDC is two-way isentropic mixing, which climatologically increases subtropical O3S tendencies above and south of the subtropical jet while reducing extratropical O3S tendencies throughout the stratosphere (Fig. 7a-c). In the tropical lower stratosphere (~80 hPa), tendencies peak during summer in present day analyses (Abalos et al. 2013) and in the preindustrial control climatology (Fig. 7c). RCP8.5 forcing generally reinforces the climatological two-way isentropic mixing in the stratosphere during each season, increasing subtropical tendencies and reducing extratropical tendencies (Fig. 7d-f). Additionally, enhanced cross tropopause mixing by eddies increases upper tropospheric O3S tendencies from 30-60N, with stronger signals during summer than winter. These anomalies are primarily associated with the RCP8.5 SSTs (Fig. 7g-i). Hardly any part of the two-way isentropic mixing responses to RCP8.5 GHGs are statistically significant (Fig. 7j-l).





## 4 Conclusions

We use three interactive chemistry WACCM experiments to analyze how stratosphere-to-troposphere transport of ozone over western North America during late winter, spring, and summer responds to worst case scenario RCP8.5 climate change during the end of the century. Lower tropospheric O3S concentrations increase up to 39% over western North America in response to RCP8.5 forcing, particularly during late winter with progressively weaker increases during spring and summer. Between the RCP8.5 GHGs and RCP8.5 SSTs, the GHGs are found to be primarily responsible for increase in lower tropospheric O3S over western North America and across the northern hemisphere.

Because lower stratospheric ozone mixing ratios are positively correlated with the amount of ozone contained in intrusions that transport mass into the troposphere (Ordóñez et al. 2007; Hess and Zbinden 2013; Neu et al. 2014; Albers et al. 2016; 2018), we document the processes modifying future lower stratospheric ozone. Considering the response to RCP8.5 GHGs, the higher ozone mixing ratios throughout the extratropical stratosphere can be attributed primarily to enhanced production associated with stratospheric cooling and higher methane concentrations (Morgenstern et al. 2018; Winterstein et al. 2019), with additional effects also from reduced ODSs (Dietmüller et al. 2021) and higher nitrous oxide concentrations (Revell et al. 2012; Butler et al. 2016). In agreement with Oberländer et al. (2013) and Chrysanthou et al. (2020), we find that the effect of the RCP8.5 GHGs on the residual mean mass streamfunction is concentrated in the upper stratosphere, manifesting as an acceleration of the BDC's deep branch, which promotes ozone transport downward primarily at high latitudes poleward of 60°N during boreal winter and spring (Fig. 6j-k). A limitation of our approach to lump all of the chemical effects of reduced ODSs, methane, nitrous oxide, and other species on ozone production into one "response to RCP8.5 GHGs" is that we cannot identify which chemical pathways are primarily responsible for the enhanced lower stratospheric ozone production. Even with the appropriate WACCM sensitivity experiments, there is considerable spread amongst climate models regarding, for instance, whether or not reduced ODSs or future GHGs promote more future ozone production in the lower stratosphere (Dietmüller et al. 2021); moreover, our results are also surely sensitive to choice of climate change scenario and model, as future STT exhibits large inter-scenario and inter-model spread (Young et al. 2013; Morgenstern et al. 2018). Contrary to the changes to residual advective transport, we find that the majority of the two-way isentropic mixing response to the RCP8.5 GHGs is not statistically significant.

The RCP8.5 SSTs promote weak (relative to the RCP8.5 GHGs) scattered regional increases and decreases in lower tropospheric O3S. Over the North Pacific, the lower tropospheric O3S increases are co-located with the low pressure center belonging to the largest trough of a tropics-extratropics planetary scale wave that forms over the North Pacific, similar to the PNA wavetrain, in response to the RCP8.5 SSTs. When the amplitude of this wave is largest during late winter, O3S increases by nearly 400 ppb within the wave's largest trough at 200 hPa, a doubling of O3S relative to the preindustrial control climatology. A large part of this trough is located in the lower stratosphere at 200 hPa, illustrating that planetary waves can





introduce high amplitude zonal asymmetries into the lower stratospheric ozone "reservoir" that then coincide with regionally
enhanced STT. In agreement with Reed (1950), we attribute the co-location between lower stratospheric troughs (ridges) and
enhanced (reduced) ozone to horizontal advection and vertical motion induced by the North Pacific planetary scale wave.
Although their studies focus on ENSO, Zhang et al. (2015) and Albers et al. (2022) each provide more detailed observational
and model-based evidence in favor of this physical mechanism.

To control for the natural month-to-month fluctuations in stratosphere-to-troposphere transport over western North America
that arise in association with the spring transition of the North Pacific jet, in all of our analyses, all results are presented as a
function of the three phases (late winter, spring, and summer) of the North Pacific jet as they are defined in Breeden et al.
(2021). The timing of the spring transition in the WACCM preindustrial control is found to be consistent with reanalysis and
we find that the model simulates a peak in lower tropospheric O3S over western North America during the preindustrial spring
transition, mimicking the seasonal peak of STT into the planetary boundary layer over western North American during this
time (Breeden et al. 2021). The RCP8.5 SSTs are found to increase the year-to-year variability of the North Pacific jet's
seasonal evolution particularly during spring and summer, broadening the distribution of days on which the spring transition
may end, which in theory could coincide with more erratic year-to-year fluctuations in STT of ozone in the future. Despite
this, we find no statistically significant change in the timing of the spring transition in response to RCP8.5 forcing. Since the
experiments use fixed repeating annual cycles of sea surface temperature and therefore by construction do not include ENSO
variability, which is known to modify the seasonal variability of the North Pacific jet (Langford 1999; Zhang et al. 2015;
Breeden et al. 2021; Albers et al. 2022) and which may itself change due to climate change, our results cannot be used to
comprehensively establish whether or not the seasonal variability of the North Pacific jet, particularly its spring transition, will
change in response to climate change. Our results do however illustrate that SSTs have a strong effect on the North Pacific jet
and in general, the RCP8.5 SSTs account for the majority of changes to the large-scale atmospheric circulation in the full
RCP8.5 forcing. The tropospheric warming associated with the RCP8.5 SSTs lifts the tropopause to higher altitudes, while
depressing the isentropes to lower altitudes. The late winter North Pacific jet accelerates, elongates, and narrows due to the
RCP8.5 SSTs. The spring and summer jets shift equatorward. The acceleration of the BDC's shallow branch, which increases
subtropical upper tropospheric O3S tendencies, is mostly accounted for by the RCP8.5 SSTs as is some of the deep branch
acceleration. In addition, the RCP8.5 SSTs account for the two-way isentropic mixing responses near ~ 50 hPa, which
reinforces the climatological patterns of mixing that enhance (reduce) subtropical (extratropical) ozone tendencies, and the
cross tropopause mixing that reinforces mid-latitude upper tropospheric O3S tendencies. Considering that the RCP8.5 SSTs
accounts for many of the changes to the large-atmospheric circulation, an avenue for future research is to analyse inter-model
spread in the future residual mean circulation response (Oman et al. 2010; Butchart 2014; Abalos et al. 2021) and the two-way
isentropic mixing response (Eichinger et al. 2019; Abalos et al. 2020) as a function of inter-model spread in future SSTs.





**Code and Data Availability**
The code used to perform this analysis can be accessed by personal communication with the corresponding author. The
WACCM simulation data used to create the figures can be accessed here:
https://csl.noaa.gov/groups/csl8/modeldata/data/Elsbury_etal_2022/

**Author Contributions**
DE wrote the code to do the analyses, created the figures, and wrote the manuscript. AHB ran the climate model experiments.
AHB, JRA, MLB, and AOL edited and provided comments on the manuscript.

**Competing Interest**
The authors declare no conflicts of interest.

**Financial support**
John R. Albers and Dillon Elsbury were funded in part by National Science Foundation grant #1756958.





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
