# Peer review of "The response of the North Pacific jet and stratosphere-to-troposphere transport of ozone over western North America to RCP8.5 climate forcing"

_Atmospheric Chemistry and Physics, 2022_

## Author Response (AR1)

**Note that the line numbers where the relevant changes to the manuscript text are made are written before each comment. Sorry! These line numbers were not included in the interactive discussion reviewer responses.**

Thank you for the helpful reviews, which have hopefully improved the clarity and quality of this manuscript. There are a few changes that we have made, which must be stated for both reviewers and the editor. An issue pointed out by both reviewers was that our discussion of the role of ozone depleting substances (ODSs) in the conclusion was misleading (it was). We have added text to the methods and conclusion sections to clarify that the ODS concentrations are higher in the RCP8.5 experiment (EXP2) compared to the preindustrial control experiment (EXP1) and we now discuss the implications of this on the ozone responses that we present.

Previously, since one of our experiments (EXP3) was referred to as the "RCP8.5 SST" experiment and some of our responses were referred to as "RCP8.5 SST" responses, it was unclear when we were referencing an experiment or a response to an experiment. To clear this up, now in each case when we are mentioning an experiment, the word "experiment" is used explicitly along with the label for the experiment in parentheses (see Table 1: EXP1, EXP2, or EXP3). Conversely, as we decompose the atmospheric response to full RCP8.5 forcing into contributions from RCP8.5 SSTs and RCP8.5 GHGs, we now refer to these constituent responses exclusively by saying "SSTs alone" or "GHGs alone." The figure labels and captions have been updated accordingly.

Following Figure 4, the previous version of this manuscript had a paragraph expressing that there is a duality between the stationary wave response to RCP8.5 conditions and the acceleration of the Brewer Dobson Circulation that may be inferred from our Figure 4. This paragraph has been removed. While not shown, the stationary wave response to full RCP8.5 forcing is more complex than was appreciated during the previous manuscript submission and realizing this rendered that since deleted paragraph questionable.

The article examines the response of the stratosphere-to-troposphere transport of ozone to a high-emissions climate change scenario (RCP8.5) using a set of time-slice simulations with WACCM. The focus is over North America and the spring transition of the Pacific jet following previous works, and the response to GHG and SST are separated. The results show increased ozone transport into the troposphere in that region peaking in late winter. The paper is well written and results are clearly presented and adequately discussed. However, there are some important issues that need to be discussed before the paper can be published, which I listed below as major comments.

Major comments:

- Figure 2 shows that the increase in ozone transport into the troposphere is dominated by GHG. This is in contrast with the results in the rest of the paper, which demonstrate a fundamental role of the SST for O3S in the lower stratosphere and upper troposphere (UTLS). How should we interpret this difference? Does it imply that changes in the lower stratospheric reservoir do not

translate into tropospheric ozone, contrary to what is typically considered? Or does it reflect a disconnection between UTLS and the deep intrusions reaching the middle-lower troposphere? Or something else? This should be discussed in the manuscript.

The relevant changes to the manuscript are made in the conclusion (L458-463).

Changes in the lower stratospheric O3S reservoir do translate into tropospheric O3S, however in the case of the SSTs alone, this change is highly regional. The regions at 700 hPa in which O3S increases in response to the SSTs alone are co-located, but beneath, the regions of the UTLS that see O3S increases near 200 hPa (Fig. 2 vs. Fig. 4). Similarly, the quasi-zonally symmetric 700 hPa O3S response to the GHGs alone resembles the quasi-zonally symmetric 200 hPa O3S response to the GHGs alone. This suggests that spatial composition of the lower tropospheric O3S response tends to mirror the UTLS O3S spatial composition. Similar results are shown in Albers et al. (2022, Figs. 4&8), which shows that the zonally asymmetric upper tropospheric (200-400 hPa) O3S increases and decreases are positioned above and poleward of the 800 hPa O3S increases and decreases. This has now been mentioned in the conclusion.

There does not appear to be a disconnection between the UTLS and intrusions reaching into the middle-lower troposphere. To approximate the number of UTLS wave breaking events, the negative meridional potential vorticity gradient occurrence frequency was calculated at every grid point along the 345 Kelvin isentrope. The SSTs alone account for the majority of the full RCP8.5 wave breaking response. The SSTs alone increase wave breaking equatorward and poleward of the North Pacific jet during winter, over east Asia and from the central-eastern Pacific to the Atlantic during spring, and with increases across North America during summer (Fig. R1g-i). However, the 345 Kelvin O3S response is regional and sometimes negative, so even if there are increases in wave breaking (and presumably more stratospheric intrusions), in some cases this will coincide with O3S depleted air entering the troposphere. The 700 hPa O3S responses to the SSTs alone shown in the manuscript reflect this. The GHGs alone on the other hand do not modify wave breaking frequency (Figs. R1j-l). This may suggest that just having more O3S in the lower stratosphere and having the climatological number of wave breaking events (Figs. R1a-c) is sufficient to increase lower tropospheric O3S.

Albers, J. R., Butler, A. H., Langford, A. O., Elsbury, D., & Breeden, M. L. (2022). Dynamics of ENSO-driven stratosphere-to-troposphere transport of ozone over North America. *Atmospheric Chemistry and Physics*, *22*(19), 13035-13048.

**345 K O3S (ppb) & RWB (#)**

[Figure]

Figure R1: 345 Kelvin O3S (shaded, ppb) and Rossby wave breaking occurrence frequency (contours, +/- 1 event) responses to RCP8.5 boundary conditions. (a-c) show the preindustrial climatologies of O3S in alternate shading and the wave breaking occurrence frequency in contours. (d-f) show responses to RCP8.5 conditions, (g-i) same, but for SSTs alone, and (j-l) same, but for GHGs alone. The phases of the jet are shown in successive columns. The preindustrial control dynamical tropopauses (PV = 2 PVU) for each season are shown in blue and anomalous tropopauses are shown in cyan.

- Figure 5 suggests that GHG enhance substantially the ozone concentrations in the extratropical lower stratosphere reservoir, but then in Figures 6 and 7 the GHG have no effect. Does this mean that the ozone local production (rather than the transport by the BDC) is enhanced with GHG increase? If so, it seems quite an important point to make, given that the ozone enhancement is larger than that due to SST.

The relevant changes to the manuscript are made in the conclusion (L429-444).

This is an important point and the role of chemical production is probably not emphasized enough in the previous manuscript. Ozone production in response to the GHGs alone is important and this can be discerned by comparing Figure 5 with Figures 6 and 7. Since the transport pathways (as shown in Figs. 6 and 7) cannot explain the extratropical lower stratospheric O3S increases, net production of O3S must be important. This is now emphasized more in the conclusion.

- The lower stratospheric ozone reservoir changes show a pattern that resemble the stationary wave response to a positive ENSO phase SST anomaly. As stated in the paper, other models

could produce different SST patterns leading to different stationary waves and thus ozone changes. Nevertheless, the increase in ozone STT is consistent across models. What does this mean for the role of the ozone wave-like anomalies on the zonal mean trends? The answer could go in the direction of the response to zonal mean SST dominating over the response to SST zonal anomalies, as found in Chrysanthou et al. (2020). A discussion on this point would improve the paper.

This is an interesting question, which we can only speculate on rather than definitively answering. Ozone-wave-like anomalies are likely important for zonal mean trends. Chrysanthou et al. show that zonally *symmetric* SSTs coincide with a large enhancement of the stratospheric upward Eliassen-Palm (EP) flux. Conversely, they show that the vertical EP flux response to the zonally *asymmetric* SSTs is comparatively weak (their Fig. 7). This may lead one to say that the zonally symmetric processes dominate zonal mean ozone trends, but this cannot be true. The vertical EP flux is proportional to the eddy heat flux (v'T', primes denoting deviations from the zonal mean) and the eddy heat flux is related to the structure of the stationary wave (long-term zonal mean geopotential height removed from full geopotential height field) (e.g., Lubis et al. 2016). For the vertical EP flux to increase in response to zonally symmetric SSTs as Chrysanthou et al. show, this probably means (1) that there is an anomalous stationary wave response to the zonally symmetric SSTs and (2) that this anomalous stationary wave response projects favorably onto the climatological stationary wave, constructively interfering with it (e.g., Fletcher and Kushner 2011), thereby enhancing upward EP flux. Should (1) and (2) be correct, then this anomalous stationary wave would likely modify ~ 200 hPa O3 in a similar way as we show in this manuscript, supporting the hypothesis that ozone-wave-like anomalies are important for zonal mean trends. In agreement with this, Oberlander et al. (2013, Fig. 8) show that the stationary wave response to climate change controls the high latitude (70N) 70 hPa residual mass streamfunction change, hence there is a duality between ozone-wave-like anomalies and the BDC. Chrystanthou et al.'s result that uniform SST warming forces a stronger upward wave response than the zonally asymmetric SSTs is interesting and worth better understanding.

The opposing stance that ozone-wave-like anomalies are not important for zonal mean trends seems difficult to justify. Although the BDC is zonally symmetric, it's dependent on wave (deviations from zonal symmetry) interactions with the mean flow. These interactions with the mean flow are regional (e.g., Takaya and Nakamura 2001, Fig. 2) and recent literature is pointing to the residual mean meridional circulation response to non-conservative (in the sense that non-accelerations conditions are not met) wave mean flow interactions being regional too (Sato et al. 2021). However, chemical changes associated with well mixed greenhouse gasses may be regarded as a quasi zonally symmetric perturbation to the atmosphere. The CO2 increase in response to climate change, for instance, modifies the BDC and leads to production of ozone via stratospheric cooling. N2O, CH4, and ODSs may be regarded as well mixed as well and their concentrations are certainly important for zonal mean ozone trends. The ozone changes associated with chemistry (e.g., CO2, N2O, CH4, and ODSs) are undoubtedly

important and we do not expect these chemical changes to be associated with ozone-wave-like anomalies near the UTLS in any obvious way.

Chrysanthou, A., Maycock, A. C., & Chipperfield, M. P. (2020). Decomposing the response of the stratospheric Brewer–Dobson circulation to an abrupt quadrupling in CO 2. *Weather and Climate Dynamics*, *1*(1), 155-174.

Fletcher, C. G., & Kushner, P. J. (2011). The role of linear interference in the annular mode response to tropical SST forcing. *Journal of Climate*, *24*(3), 778-794.

Lubis, S. W., Matthes, K., Omrani, N. E., Harnik, N., & Wahl, S. (2016). Influence of the quasi-biennial oscillation and sea surface temperature variability on downward wave coupling in the Northern Hemisphere. *Journal of the Atmospheric Sciences*, *73*(5), 1943-1965.

Oberländer, S., Langematz, U., & Meul, S. (2013). Unraveling impact factors for future changes in the Brewer‐Dobson circulation. *Journal of Geophysical Research: Atmospheres*, *118*(18), 10-296.

Sato, K., Kinoshita, T., Matsushita, Y., & Kohma, M. (2022). A new three-dimensional residual flow theory and its application to Brewer–Dobson circulation in the middle and upper stratosphere. *Journal of the Atmospheric Sciences*, *79*(2), 429-448.

Takaya, K., & Nakamura, H. (2001). A formulation of a phase-independent wave-activity flux for stationary and migratory quasigeostrophic eddies on a zonally varying basic flow. *Journal of the Atmospheric Sciences*, *58*(6), 608-627.

- In lines 105-107 the ODS are said to increase from the pre-industrial to the RCP8.5 end-of-century simulations (note that it would be convenient to state here by how much they do increase in order to quantify their effect). However, in Line 437 it is stated "with additional effects from reduced ODS". I suspect this confusion is due to the different simulations considered in Dietmuller et al. (2021) versus this manuscript. Please, clarify what is the role of ODS if any in your analysis. Also, even though the role of ODS is not explicitly examined in this work given the comparison to pre-industrial conditions, previous studies have shown that ODS decline is the dominant forcing of global ozone STT increase over the century (Banerjee et al. 2016, Meul et al. 2018, Abalos et al. 2020). This is an important point that should be clearly stated in the paper in order to avoid confusion.

The relevant changes to the manuscript are made in the conclusion (L429-444).

As you guessed, this was an oversight on our part while trying to relate Dietmuller et al. (2021)'s results to our own. Their study focuses on 1990s-2100 whereas ours focuses on preindustrial vs late 21st century and this important difference was forgotten at the time of writing.

This comment did lead us to better appreciate the role of the ODSs in our runs and text has been added to the methods section and conclusions to reflect the importance of the ODSs.

Dietmüller, S., Garny, H., Eichinger, R., & Ball, W. T. (2021). Analysis of recent lower-stratospheric ozone trends in chemistry climate models. *Atmospheric Chemistry and Physics*, *21*(9), 6811-6837.

Minor comments:

- L88: Please specify how stratospheric ozone loss is treated in the troposphere. There are different ways in which this has been made and it is beneficial to explicitly include this information.

The updated manuscript text is from L89-L91.

This update has been made. The loss in the troposphere is explicitly calculated and includes loss due to photolysis and chemistry, plus dry deposition. This appears to be the same methodology used by Abalos et al. (2020).

Abalos, M., Orbe, C., Kinnison, D. E., Plummer, D., Oman, L. D., Jöckel, P., ... & Dameris, M. (2020). Future trends in stratosphere-to-troposphere transport in CCMI models. *Atmospheric chemistry and physics*, *20*(11), 6883-6901.

- L92-93: "to remove interannual variability driven by the ocean (e.g. variability due to ENSO)" It is true that there is no variability in your experiments. However, a clear ENSO signal shows up in the climatology of the SST change, presumably due to more frequent/intense events. So you are not really removing this effect (and indeed you refer to it later on). So I suggest rephrasing this.

The updated manuscript text is from L93-L97.

True, thanks for this point. We have updated the text to emphasize that "interannual fluctuations in SSTs" are not present in these experiments rather than saying that intimating that ENSO variability is not present when, as you point out, it is.

To keep the text as simple as possible, we now write "Interannual SST fluctuations (which may for instance arise due to ENSO) are excluded from our experiments, hence, our results cannot comprehensively establish how RCP8.5 forcing modifies the timing of the spring transition."

For the purposes of the review, we cannot be completely sure that the El Nino like signature prescribed in the RCP8.5 SSTs is responsible for non statistically significant changes in North Pacific jet variability. In this study, RCP8.5 conditions make the largest impact on the North Pacific jet's variability during spring and summer. However, in reanalysis, ENSO has its biggest effect on the Pacific jet's variability during late winter (Breeden et al. 2021, Fig. 8). Since the changes in RCP8.5 jet timing cannot easily be explained by the El Nino like signature in our prescribed SSTs, this could suggest that extratropical SSTs have a strong effect on the jet's variability too. We do not know though. Our understanding of which portion (El Nino vs tropical vs extratropical vs Indian Ocean) of the SST field influences the jet variability is too elementary at this time so we choose to omit further discussion on this point in the manuscript.

- L121-124 seem in contradiction with L117-120. Please clarify what you mean.

The updated manuscript text is from L128-L132.

Thanks for pointing this out. We have removed this text and replaced it with:

Note that we derive our response to GHGs alone as the residual between EXP3, which includes RCP8.5 SSTs only, and EXP2, which includes full RCP8.5 forcing. If the SST forcing and GHG forcings interact non-linearly, the response to GHGs alone as we define it (EXP2 - EXP3) may be different from the response to GHGs that could be obtained by comparing a preindustrial experiment to an experiment with RCP8.5 greenhouse gases and SSTs fixed to 1850 conditions. The additivity of the response to RCP8.5 SSTs alone and the response to RCP8.5 GHGs alone will have to be assessed in future work.

- L241: Is this robust also across climate models?

Yes! Consider Harvey et al. (2020) for instance, their Fig. 3, which shows the future minus present day multi-model 250 hPa zonal wind anomalies. The multi-model jet response they show resembles what we present in Figure 2.

Harvey, B. J., Cook, P., Shaffrey, L. C., & Schiemann, R. (2020). The response of the northern hemisphere storm tracks and jet streams to climate change in the CMIP3, CMIP5, and CMIP6 climate models. *Journal of Geophysical Research: Atmospheres*, *125*(23), e2020JD032701.

- L253: Is the amplified surface warming related to changes in sea ice? I assume sea ice is also imposed as a boundary condition?

Yes, sea ice is imposed as a boundary condition along with the SSTs. The rather large tropospheric temperature response over the Arctic (presumably due to the ice-albedo feedback) can be seen in the supplementary SST alone temperature response.

- Fig. 4: Top row: are there negative values in low latitudes or is it an effect of the colorbar? What are the units of O3S? Caption: "stationary wave (contou*r*s, long-term zonal...) ". Stationary waves are not a physical magnitude, change to something like "Stationary waves visualized by geopotential height deviation from the long-term mean zonal mean (contours, in meters)"

The top row colorbar has been updated to more easily show that the 200 hPa O3S values do not reach zero ppb anywhere visible on Fig. 4a-c.

The units for O3S are in the title of the figure and we have added them to the caption as well.

We clarified what the contours overlaid on Figure 4 show.

- L312: Why is the modified flow implied by the wave phase tilt?

Following further analysis, the structure of RCP8.5 stationary wave response is more complex than was thought at the time of writing. As a result, this entire paragraph (previously L306-316), which speculated on the relationship between the stationary wave response and the BDC, has been removed.

For completeness, we have still answered your questions below:

The wave's tilt tells us where it's propagating upward and downward. Westward tilt with increasing height means upward propagation and eastward tilt with height means downward propagation (e.g., Harnik and Lindzen 2001). This can be inferred from the high latitude (60N-70N) December climatological stationary wave (left) and the December-January Plumb (1985) wave activity flux (right) (vertical and zonal components shows as vectors and 3D divergence shown as contours w/+/- 0.8 sigma contours). **Note that this plot is cropped from a larger figure, which is why the left plot is for December and the right plot is for DJ; the stationary wave, viewed either in eddy height anomalies or in wave activity fluxes, looks very similar between December and January.** For the right plot, all Plumb flux components (including the 3D divergence) have been normalized by their climatological standard deviation just to make the plotting easy. The sign of the vectors is what is important and it's shown that the upward/downward vectors mirror the westward/eastward phase tilt with height.

[Figure]

**December - January (DJ) stationary wave 60°N-70°N GEOPx (m)**

Regarding the modified flow, if we only consider the vertical component of the EP flux or Plumb flux, then if its vertical derivative varies at all with height (it generally does in my experience), there will be EP flux divergence or convergence. See Fig. 2 of Dunn-Sigouin and Shaw (2015).

Harnik, N., & Lindzen, R. S. (2001). The effect of reflecting surfaces on the vertical structure and variability of stratospheric planetary waves. *Journal of the atmospheric sciences*, *58*(19), 2872-2894.

Plumb, R. A. (1985). On the three-dimensional propagation of stationary waves. *Journal of Atmospheric Sciences*, *42*(3), 217-229.

Dunn-Sigouin, E., & Shaw, T. A. (2015). Comparing and contrasting extreme stratospheric events, including their coupling to the tropospheric circulation. *Journal of Geophysical Research: Atmospheres*, *120*(4), 1374-1390.

Technical:

- L167: "distribution of median dates" : remove "median"

The updated manuscript text is from L168-L182.

Done.

- Fig. 2 caption: What does the box show?

The updated manuscript text is from L221-L222.

Text added to caption.

- L122: "still coincide with roughly 10-35%" → "still imply roughly a 10-35%"

The updated manuscript text is on L228.

Done.

- L259: "purely chemical changes in the atmosphere" Strictly, this applies to all climate change impacts, do you mean chemical changes in the *stratosphere*?

The updated manuscript text is on L266.

Yes, that is what was meant.

 L316: Fig. 4k → Fig. 4g

The paragraph containing this text has been removed.

- L334: 235ºE-260ºE: does this correspond to the box in Fig. 2? If so, please add it.

The updated manuscript text is on L330.

Yes, this does correspond to the Fig. 2 box. Added text noting this.

- L389: "=" should be ")"

The updated manuscript text is on L386.

Thanks for spotting this, fixed!

########################### Reviewer 2 ###############################

The paper by Elsbury et al. deals with the response of ozone STT and North Pacific jet to RCP8.5 forcing over the western North America region which is a well-known hot spot of ozone STT. This is a very interesting paper on the main mechanisms of future ozone STT over western North America. It is well written and structured, and the results are nicely presented. My only concern is the terminology of the experiments and the associated discussion which I find misleading. I would be happy to suggest publication of the manuscript in ACP once my comments below are addressed:
Comments:

1. The future SSTs mainly result from the future GHGs. What the authors present is a decomposition of the overall GHGs effect in that of SSTs and the residual of GHGs (including also the ODSs) which to my understanding is mainly the chemical effect. The EXP2 described with term RCP8.5 is the one that represents the whole GHGs effect. Therefore, using the terms "RCP8.5 SSTs" and "RCP8.5 GHGs" is somehow misleading as both constitute the overall GHGs effect (term "RCP8.5"). The same applies to the corresponding discussion.

The most relevant changes to the manuscript are made in the methods (L114-132). Changes in labeling (e.g., "SSTs alone") are made throughout the entire manuscript.

Thank you for pointing this out. We updated our methods section to emphasize that we are decomposing the full RCP8.5 forcing (GHGs + ODSs + SSTs) into a GHG portion, which includes the ODSs, and an SST portion. We now emphasize that the SST changes are themselves  induced by the GHG changes. We also now emphasize that the ODSs are part of our "response to the GHGs alone" and discuss the implications of this in the methods.

In the previous manuscript, referring to the experiments and to the anomalies using the same names, e.g., "RCP8.5 SST," was confusing. Following Oberlander et al. (2013) and Chrysanthou et al. (2022) whose experimental setups were similar to ours, we now reference the experiments as "EXP1, EXP2, and EXP3" or in some cases, as "the preindustrial control experiment" or the "RCP8.5 experiment." Throughout the result section, the cumbersome references to anomalies of "RCP8.5 SSTs" and "RCP8.5 GHGs" have been removed and are now more simply referred to as the "SSTs alone" or it's simply said "The SSTs alone do blank.."

Chrysanthou, A., Maycock, A. C., & Chipperfield, M. P. (2020). Decomposing the response of the stratospheric Brewer–Dobson circulation to an abrupt quadrupling in CO 2. *Weather and Climate Dynamics*, *1*(1), 155-174.

Oberländer, S., Langematz, U., & Meul, S. (2013). Unraveling impact factors for future changes in the Brewer‐Dobson circulation. *Journal of Geophysical Research: Atmospheres*, *118*(18), 10-296.

2. I find the term "RCP8.5 GHGs" misleading, as this also includes the ODSs impact on future ozone STT. Although, the authors nicely discuss the limitations of their approach in

the conclusions, I feel that the term "RCP8.5 GHGs" gives the impression that ODSs are not included, and someone must specifically find that information (that are included) in the text.

The relevant changes to the manuscript are made in the methods (L108-126) and further mentioned in the conclusion (L442-L444).

This is an important point and we have added text to methods and to the conclusions to explicitly state that the ODSs are included as part of the GHG response. The implications of binning the CH4, CO2, N2O, and ODS responses together is now discussed as well.

3. Moreover, I am a bit puzzled with how ODSs are treated in the simulations. The authors state (P4, L105-107) "There are also increased concentrations of ozone-depleting substances (ODS; e.g., chlorofluorocarbons) relative to the preindustrial experiment, due to the long lifetimes of these substances, which were emitted prior to the Montreal Protocol.", but in the Conclusions they attribute higher ozone concentrations also "…from reduced ODSs" (P19, L437). Is the latter statement referring to a comparison with respect to the present period? If yes, the comparison presented is relative to the preindustrial period. Please clarify this.

The relevant changes to the manuscript are made in the methods (L114-126) and further mentioned in the conclusion (L442-L444).

Thank you for spotting this contradiction. This sentence, which was previously in the conclusion, was referencing the results of Dietmuller et al. (2021), whose study focuses on 1990s-2100. However, this was misleading because as you point out, our study focuses on preindustrial vs. late 21st century. Following this comment and a similar comment from the other reviewer, we have added text to the methods section and to the conclusion to better clarify the role of ODSs in these simulations.

Dietmüller, S., Garny, H., Eichinger, R., & Ball, W. T. (2021). Analysis of recent lower-stratospheric ozone trends in chemistry climate models. *Atmospheric Chemistry and Physics*, *21*(9), 6811-6837.

Minor Comments:

L32-L33:  Recently Zanis et al. (2022) studied the climate change penalty and benefit on ozone air-quality using simulations from CMIP6 ESMs. I believe this reference can be included here.

This addition is on L33.

Added!

L82: What is the resolution of WACCM model near the tropopause?

Horizontally, 1.9 degrees latitude by 2.5 degrees longitude. If you are referencing the vertical resolution, it ranges between 1.1 and 1.4 kilometers throughout the lower stratosphere. In a sense, the vertical resolution near the tropopause is much finer. The finite volume dynamical core uses fluctuating vertical coordinates: the material surfaces that bound the finite volumes, which contain some amount of tracer, can be compressed or spread apart (Neale et al. 2010). The material surfaces are then remapped back to the predefined model levels, and transport proceeds by considering the pressure difference between the current (fluctuating material surfaces) finite volume and the predefined (fixed vertical coordinate) finite volume (Abalos et al. 2013).

Neale, R. B., Chen, C. C., Gettelman, A., Lauritzen, P. H., Park, S., Williamson, D. L., ... & Taylor, M. A. (2010). Description of the NCAR community atmosphere model (CAM 5.0). *NCAR Tech. Note NCAR/TN-486+ STR*, *1*(1), 1-12.

Abalos, M., Randel, W. J., Kinnison, D. E., & Serrano, E. (2013). Quantifying tracer transport in the tropical lower stratosphere using WACCM. *Atmospheric Chemistry and Physics*, *13*(21), 10591-10607.

L88: How is tropopause defined in WACCM?

WACCM uses the lapse rate tropopause. At high latitudes ( >55 degrees), in the event that the tropopause cannot be found, the model will use the climatological tropopause.

L218: Please include degree symbols (and where else applicable).

This change is on L226-227.

Added!

L271: Maybe "The 200 hPa O3S".

This change is on L278.

Added!

References

Zanis, P., D. Akritidis, S. Turnock, V. Naik, S. Szopa, A.K. Georgoulias, S.E. Bauer, M. Deushi, L.W Horowitz, J. Keeble, P. Le Sager, F.M. O'Connor, N. Oshima, K. Tsigaridis, and T. van Noije, 2022: Climate change penalty and benefit on surface ozone: A global perspective based on CMIP6 earth system models. Environ. Res. Lett., 17, no. 2, 024014, doi:10.1088/1748-9326/ac4a34.